# Comparative Study of Onion-like Carbons Prepared from Different Synthesis Routes towards Li-Ion Capacitor Application

Antonius Dimas Chandra Permana [1], Ling Ding [1], Ignacio Guillermo Gonzalez-Martinez [1], Martin Hantusch [1], Kornelius Nielsch [1,2], Daria Mikhailova [1,*] and Ahmad Omar [1,*]

1   Leibniz Institute for Solid State and Materials Research (IFW) Dresden e.V., Helmholtzstraße 20, 01069 Dresden, Germany
2   Institute of Materials Science, Technische Universität Dresden, Helmholtzstraße 10, 01069 Dresden, Germany
*   Correspondence: d.mikhailova@ifw-dresden.de (D.M.); a.omar@ifw-dresden.de (A.O.)

**Abstract:** Li-ion capacitors (LIC) have emerged as a promising hybrid energy storage system in response to increasing energy demands. However, to achieve excellent LIC performance at high rates, along with cycling stability, an alternative anode to graphite is needed. Porous high-surface-area carbons, such as onion-like carbons (OLCs), have been recently found to hold high potential as high-rate-capable LIC anodes. However, a systematic understanding of their synthesis route and morphology is lacking. In this study, OLCs prepared from self-made metal organic frameworks (MOFs) Fe-BTC and Fe-MIL100 by a simple pyrolysis method were compared to OLCs obtained via high-temperature annealing of nanodiamonds. The LICs with OLCs produced from Fe-BTC achieved a maximum energy density of 243 Wh kg$^{-1}$ and a power density of 20,149 W kg$^{-1}$. Furthermore, excellent capacitance retention of 78% after 10,000 cycles was demonstrated. LICs with MOF-derived OLCs surpassed the energy and power density of LICs with nanodiamond-derived OLCs. We determined the impact of the MOF precursor structure and morphology on the resulting OLC properties, as well as on the electrochemical performance. Thus, MOF-derived OLCs offer significant potential toward high-performance anode material for LICs, enabling control over structure and morphology, as well as easy scalability for industrial implementation.

**Keywords:** onion-like carbon; metal organic framework; Li-ion capacitor; high energy density; high power density

## 1. Introduction

Li-ion capacitors (LICs), which constitute a bridge between a Li-ion batteries (LIBs) and supercapacitors, represent a promising electrochemical energy storage technology. Utilizing the anode from LIBs, wherein bulk interaction/diffusion is the storage mechanism, results in high energy density. On the other hand, when combined with a supercapacitor electrode, double-layer capacitance allows for rapid Li-ion charge–discharge, demonstrating high power density and stable long-term cycling.

In order to achieve optimal results for an LIC system, pairing the best features of the LIB anode and the supercapacitor cathode is the key. Activated carbon, with its large specific surface area and high conductivity, is a well-established supercapacitor electrode and is also used effectively as an LIC cathode [1]. A wide variety of LIB anodes is available, where the material properties and charge storage mechanism(s) vary significantly, such as carbon nanotubes [2] (single-walled [3] and multi-walled [4]), etc. Anode materials can be separated into two major classes: organic and inorganic compounds. Organic or carbonaceous materials, in particular, are known to reach substantial energy density and can satisfy cost and environmental concerns. Graphite, one of the most common anode materials for LIBs, has a low lithiation potential (~0 V vs. Li/Li$^+$), a relatively high theoretical capacity (372 mAh g$^{-1}$), and high cycling reversibility [5]. However,

graphite is subject to limitations with respect to Li-ion diffusivity in the material during the intercalation/deintercalation process, affecting its high rate performance [5]. Moreover, issues of lithium plating, as well as dendritic growth on the surface of graphite, cannot be neglected in terms of safety [6]. Owing to associated challenges, a limited number of carbonaceous LIB anodes have been used in LIC systems [7,8].

Onion-like carbon (OLC), a novel carbon allotrope, was recently introduced as a promising alternative anode material for LIBs and LICs [9]. Owing to its unique structure of concentric carbon shells, OLC material possesses high electrical conductivity, a large surface area, and a graphitic structure with high defect concentration [10]. Consequently, the potential of OLC has been explored as an electrode material and conductive additive for supercapacitors and batteries [10]. OLCs have also been tested for application in photovoltaic cells [11–14], as well as $H_2$ storage devices [15]. For further modifications, OLC can be doped with N/S [16,17], with the potential for chemical functionalization of the outer surface [18,19]. Hence, OLC material is very attractive for use in LIC technology.

A strong control of the morphology and structure of OLC material is therefore needed to achieve optimization for desired LIC applications. The structure and morphology of OLCs strongly depend on the synthesis procedure. In addition to the impact of synthesis on surface area and porosity of OLCs, the core that builds its multilayer concentric graphitic shells may either be hollow or solid. Production of OLCs via nanodiamond annealing is the most common route [10]. Nanodiamond-derived OLCs have a surface area up to 600 m$^2$ g$^{-1}$, leading to a satisfactory capacitive performance [10]. However, OLCs produced via this route have solid cores and highly ordered shells due to a high annealing temperature [10], affecting their electrochemical capabilities [20]. Still, nanodiamond-derived OLCs were used as metal-oxide-hybrid templates for LIC applications [21–23]. Another commercially viable synthesis route for OLCs is through carbonization of natural waste material, although this process suffers from issue of reproducibility and small surface areas (~140 m$^2$ g$^{-1}$ [24] and ~50 m$^2$ g$^{-1}$ [25]), limiting electrochemical performance for supercapacitors and hybrid capacitors.

Recently, another alternative route for OLC synthesis was demonstrated through pyrolysis of metal-organic frameworks (MOFs) [26]. OLCs produced by this method are reported to have a high surface area [9], avoiding the need for further physical/chemical activation [27]. Moreover, the MOF-derived route allows for further control over system pressure and atmosphere, as well as the possibility of controlling the structural parameters and dopants, enabling desired modification of structure and morphology of the produced OLCs. Additionally, the MOF-derived synthesis of OLCs delivers high-purity products with a simple equipment setup, enabling low-cost and easy scalability [9]. Thus, this synthesis procedure is very attractive not only for the development of OLCs for improved LIC performance but also for industrial implementation. However, a systematic understanding of the parameters of MOF-derived OLC synthesis, along with their correlation to structure/morphology and LIC performance, is still nascent. Therefore, there is a need for a comparative study of OLCs involving different MOF precursors, as well as OLCs synthetized using other routes.

Herein, OLCs prepared via the MOF-derived route using two different MOFs (Fe-BTC and Fe-MIL100) are compared with OLCs prepared via the nanodiamond route. OLCs are further investigated as anode materials for LIC applications. Although it appears that MOF-derived OLCs have a different surface area and porosity, their performance in LICs are still better than that of nanodiamond-derived OLCs.

## 2. Materials and Methods

### 2.1. Materials

Fe(NO$_3$)$_3$·9H$_2$O (analytical grade, CAS: 7782-61-8; Sigma-Aldrich, St. Louis, MO, USA) and trimesic acid (1,3,5-benzenetricarboxylic acid, H$_3$BTC, 98%, CAS: 554-95-0; Sigma-Aldrich) were used as received. Fuming hydrochloric acid (HCl, 37%) and nitric acid (HNO$_3$, 65%) were purchased from Merck KGaA (Darmstadt, Germany), and ethanol was

purchased from Berkel AHK (99%; Berlin, Germany). Nanodiamond powder grade G (>87%) was purchased from PlasmaChem GmbH (Berlin, Germany). Conductive carbon black (Super C65) was obtained from Imerys (Paris, France). Activated carbon YP50F (Kuraray Chemical Co., Okayama, Japan), polyvinylidene difluoride (PVDF) powder Solef® 1013 (Solvay S.A., Brussels, Belgium), and N-methyl-2-pyrrolidone (NMP; >99.5%, Sigma Aldrich) were used as received. Additionally, 9 μm thick copper foil (MTI Corp., Richmond, CA, USA) and 15 μm thick aluminum foil (Korff AG, Oberbipp, Switzerland) were directly used as current collectors.

### 2.2. Synthesis of MOFs

Fe-BTC (iron 1,3,5-benzenetricarboxylate) was synthesized by mixing a solution of 16.3 g $Fe(NO_3)_3 \cdot 9H_2O$ with 200 mL distilled deionized (DDI) water, added dropwise into a solution of 5.7 g trimesic acid in 200 mL DDI water in an effective Fe:BTC molar ratio of 1.5:1 [28]. The solution was stirred for 15 min and washed repeatedly by DDI water until there was no more colored liquid. Washing was continued with ethanol several times until the liquid was clear. The obtained slurry was left to dry inside a fume hood at room temperature. The Fe-BTC powder was collected after grinding in a mortar.

Fe-MIL100 (Fe Materials of Institute Lavoisier index no. 100) was synthesized by mixing and grinding $Fe(NO_3)_3 \cdot 9H_2O$ and trimesic acid in a Fe:BTC molar ratio of 1.5:1 using a mortar. The mixture was transferred into an autoclave and heated to 120 °C with 4 h dwelling time in an oven [29]. The autoclave was then removed from the oven and allowed to cool down overnight. The produced slurry was washed by DDI water to remove any excess acid and dried in an oven at 60 °C overnight. The dried powder was collected, ground, and used as Fe-MIL100.

### 2.3. Synthesis of OLCs

MOF-derived OLCs were prepared according to the procedures described in a previous study [9]. An amount of 1 g of Fe-BTC or Fe-MIL100 precursor in an alumina boat was carbonized at 800 °C with 1 h dwell time in a Nabertherm tube furnace (Lilienthal, Germany) under a constant 0.15 L/min flow of argon gas through the whole process. An initial heating rate of 5 °C/min was used, and the sample was allowed to cool down overnight in the furnace. The obtained samples were ground and mixed with a solution of HCl and ethanol (3:1 *v/v*), followed by overnight stirring to remove the iron compounds. After washing with ethanol and DDI water until the waste liquid was clear, black powder was obtained by drying in air at 60 °C. The powders produced from Fe-BTC and Fe-MIL100 were labelled as OLC-B and OLC-M, respectively.

For the synthesis of OLC from nanodiamond (ND), the pyrolysis process described in [30] was adopted. To this end, 1 g of ND was placed inside a Gero tube furnace (Sheffield, UK) and heated to 1300 °C with a heating rate 5 °C/min, followed by a dwell time of 2 h, under a steady flow rate of 0.15 L/min of argon throughout the whole pyrolysis process. Thereafter, the sample was cooled down inside the tube furnace with gas flow to avoid any other reactions until room temperature was reached. After cooling down, the sample was removed from the furnace, ground, collected, and labelled as OLC-N. The described synthesis processes are presented in Figure 1.

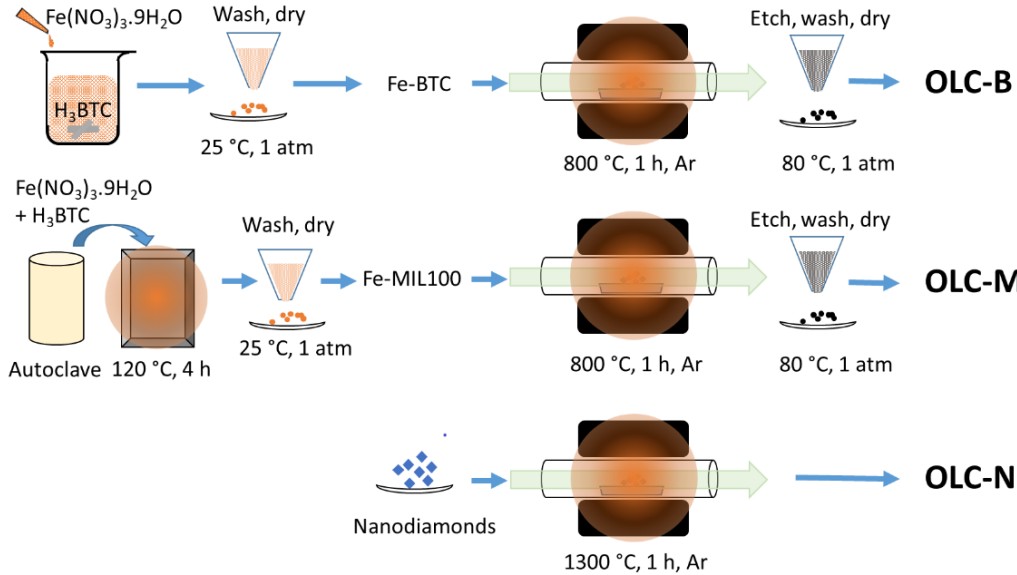

**Figure 1.** Schematic representation of synthesis processes for OLC-B, OLC-M, and OLC-N.

*2.4. Characterization*

Powder X-ray diffraction (XRD) experiments were performed in transmission mode using a STOE (Darmstadt, Germany) STADI P diffractometer with Cu K$\alpha$1 radiation ($\lambda$ = 1.5406 Å) and Co K$\alpha$1 radiation ($\lambda$ = 1.7890 Å) fitted with a curved Johann-type Ge(111) monochromator and a single-strip Dectris Mythen 1K detector (Baden, Switzerland). Samples were glued between two acetate sheets.

A Zeiss LEO Gemini 1530 microscope (Oberkochen, Germany) with an acceleration voltage of 15 kV was used for scanning electron microscopy (SEM). A FEI Tecnai F30 and a FEI-TITAN (Waltham, MA, USA) fitted with a Cs-aberration correction system operated at 300 kV were used for high-resolution transmission electron microscopy (HR-TEM).

Raman spectroscopy measurements were performed using a Thermo Scientific Smart DXR Raman spectrometer (Waltham, MA, USA) using an excitation laser wavelength of 532 nm with 3 mW laser power and a 2.1 µm spot size.

A Quantachrome Quadrasorb SI apparatus (Boynton Beach, FL, USA) was employed for nitrogen sorption measurements. Before measurements, a degassing step was performed involving a dynamic vacuum at 150 °C for 8 h. The multipoint Brunauer–Emmet–Teller (BET) method was used to calculate the specific surface areas in a relative pressure range of $0.05 \leq p/p_0 \leq 0.2$. The total pore volume was evaluated at $p/p_0 = 0.97$. Quenched solid density functional theory (QSDFT) was used to calculate the pore size distributions (PSDs) using a $N_2$-carbon equilibrium transition kernel at 77 K based on a slit-pore model.

X-ray photoelectron spectroscopy (XPS) was performed on a PHI 5600 CI spectrometer (Physical Electronics Inc., Chanhassen, MN, USA) system with Al K$\alpha$ radiation (350 W) and a hemispherical analyzer. All spectra were recorded using a pass energy of 29 eV and a step size of 0.1 eV in the energy range of 0–1000 eV. Pure metal foils were used to calibrate the binding energy scale with a binding energy (BE) of Au $4f_{7/2}$ at 84.0 eV and BE of Cu $2p_{3/2}$ at 932.7 eV. The spectra were calibrated using artificial carbon with the C 1s BE at 284.8 eV. Elemental concentrations were calculated using standard single-element sensitivity factors. Gaussian peak functions and a non-linear Shirley-type background were used to fit the spectra.

Inductively coupled plasma-optical emission spectrometry (ICP-OES) was performed on an iCAP 6500 Duo View (Thermo Fisher Scientific, Waltham, MA, USA) at three wavelengths: 240.488, 259.837, and 259.940 nm.

Fourier transform infrared (FT-IR) spectroscopy was performed using a PerkinElmer spectrometer (Waltham, MA, USA) within the wavenumber range of 500–4000 cm$^{-1}$. Thermogravi-

metric analysis (TGA) was performed using a Netzsch STA 449C (Selb, Germany) coupled with a Pfeiffer Vacuum Prisma quadrupole mass spectrometer (Aßlar, Germany). The measurements were carried out under a synthetic air atmosphere at a heating rate of 5 °C/min.

*2.5. Electrochemical Measurements*

For the anode slurry, OLC was mixed with carbon black (Super C65) as the conductive agent, with PVDF as binder in a 85:5:10 weight ratio, respectively, and NMP was used as the solvent. A Retsch MM200 shaker mill (Haan, Germany) was used to mix the slurry for 30 min at 25 Hz. The doctor blade-coating method was used to coat the prepared slurry onto Cu foil with a 200 μm blade height, using an Erichsen Coatmaster 509 film appilcator (Hemer, Germany) and a Zehntner ZUA 2000 blade (Basel, Switzerland). The coating was subsequently dried in an oven at 80 °C in air for 24 h. Electrodes with a diameter of 12 mm were punched from the coated sheet and further dried at 80 °C for at least 8 h under vacuum conditions inside a MBraun glovebox (Garching, Germany). The average active material mass loading was ~1.2 mg cm$^{-2}$ for anodes.

Similarly, for the cathode slurry, activated carbon YP50F, Super C65, and PVDF were mixed in a weight ratio of 70:10:20, with NMP as solvent. After 30 min of shaker mill mixing at 25 Hz, a blade height of 250 μm was used to coat the slurry onto aluminum foil. The weight ratio of the anode to the cathode was adjusted to ~1:3 for full-cell capacity balancing. For the cathode, the average mass loading was ~4 mg cm$^{-2}$.

Two-electrode Swagelok cells were used for electrochemical measurements of half-cell LIBs and for pre-lithiation cycling. CR2025 coin cells were used for full-cell LICs. At least 4 cells were tested under equivalent conditions to ensure reproducibility. For all cells, Celgard 2500 (Charlotte, NC, USA) was used as the separator, and for the electrolyte, LP30 (1 M LiPF$_6$ in 1:1 (*v/v*) mixture of ethylene carbonate and dimethyl carbonate (BASF, Ludwigshafen, Germany)) was used.

For LIB half-cells and pre-lithiation, 250 μm thick electrochemical-grade lithium chips (ChemPUR, Karlsruhe, Germany) were used as the counter electrode. Pre-lithiation was performed at a current density of 0.1 A g$^{-1}$ in a voltage window of $0.01 \leq U \leq 3$ V (vs. Li/Li$^+$) for 3 cycles, ending in the fully lithiated state. Then, the cells were disassembled inside the glovebox, and the electrodes were separated and immediately used to build full cells. LIC full cells were assembled with the pre-lithiated OLC electrode as anode and activated carbon (labelled as AC) as cathode.

All cell assembly was performed inside a MBraun glovebox (Garching, Germany), with H$_2$O and O$_2$ values below 0.1 ppm. Additionally, all cells were rested for 8 h before testing to ensure sufficient wetting of the electrodes and the separator with the electrolyte. A Biologic VMP3 multichannel potentiostat (Seyssinet-Pariset, France) was used for all electrochemical tests. Measurements were performed at 25 °C using a temperature chamber.

Galvanostatic cycling with potential limitation (GCPL) was undertaken at varying current rates. A voltage window of $0.01 \leq U \leq 3$ V (vs. Li/Li$^+$) was used for half-cell testing of the OLC samples. For full-cell LICs, a voltage window of $2 \leq U \leq 4$ V was applied. Cyclic voltammetry (CV) was carried out at varying scan rates, from 0.2 to 10 mV/s.

Electrochemical impedance spectroscopy (EIS) measurements were performed using an AC voltage amplitude of 5 mV in the frequency range of 200 kHz to 100 mHz. Relaxis3 software (rhd instruments GmbH & Co. KG, Darmstadt, Germany) was used for the electric equivalent circuit model simulation and to fit the EIS spectra.

The calculations for specific capacitance, specific capacity, energy density, and power density are elaborated in the Supplementary Materials. Energy density and power density were calculated according to the mass of the active material in both electrodes.

## 3. Results and Discussion

Two Fe-based MOFs were prepared with the same BTC linker, Fe-BTC and Fe-MIL100, in order to understand the impact of MOF structure on the produced OLCs. The precursor MOFs, along with commercial nanodiamond (ND), were characterized in detail. Three

OLC materials were prepared by pyrolysis of the MOFs and annealing of ND. The obtained samples were labelled as OLC-B, OLC-M, and OLC-N, corresponding to the Fe-BTC, Fe-MIL100, and ND precursor, respectively. Subsequently, detailed characterizations of the structure, morphology, and physical properties of the as-synthesized OLCs and their precursors were performed.

### 3.1. Characterization of Precursors and Produced OLCs

The precursors were characterized using XRD, TGA, and FT-IR. Figure S1a shows the XRD patterns of Fe-BTC, Fe-MIL100, and ND. Distinguishable crystalline peaks are observed for the ND sample at 43° and 76° in accordance with the literature [31]. The XRD pattern of our synthesized Fe-BTC is similar to that of commercial Fe-BTC (Basolite F300), which was used in our previous study [9]. Broad features are observed for Fe-BTC, suggesting its low crystallinity. Therefore, indexing of the XRD data was not possible. Fe-BTC is reported in the literature to have a disordered structure [32], and was previously suggested to be a derivative version of crystalline Fe-MIL100 [33]. In contrast, sharp crystalline peaks are observed for the synthesized Fe-MIL100 sample, and the corresponding planes could be indexed according to the literature [34,35]. The difference in the crystalline peaks of Fe-MIL100 as compared to Fe-BTC might be the result of exposure to heat during our synthesis procedure.

A minimal difference is observed in the FT-IR spectra for Fe-BTC and Fe-MIL100 (Figure S1b). A weaker signal at 1715 $cm^{-1}$ is observed for Fe-MIL100, representing C=O bending, and another one at 1633 $cm^{-1}$, corresponding to C=O stretching.

TGA data for Fe-BTC and Fe-MIL100 are shown in Figure S2. For Fe-BTC, ~10.6% weight loss is observed until 203 °C, which can be attributed to surface-adsorbed water molecules, as Fe-BTC synthesis does not involve heating or vacuum treatment, followed by a 10.5% loss until 315 °C, likely from the water molecules bound to iron trimers, and finally, a 48.3% loss until 426 °C due to decomposition of the BTC linker [34]. Fe-MIL100 (Figure S2b) shows no clear distinction of steps between adsorbed and bound water molecules, with a total weight loss of 16.4% until 300 °C. This is probably the result of the heat treatment during Fe-MIL100 synthesis, which can minimize the content of adsorbed water molecules. Thereafter, 40.8% weight loss is obtained due to BTC decomposition until 396 °C. Unlike Fe-BTC, an additional weight loss stage corresponding to 6.7% loss is observed. This additional step was also reported by Msahel et al. but was not discussed [36]. Overall, the iron residue after the combustion process was 33.1% for Fe-MIL100 and 28.4% for Fe-BTC. Although both Fe-BTC and Fe-MIL100 show similar chemical compositions, they differ strongly in terms of structure/disorder.

The morphology of the produced OLCs was first investigated by SEM and HR-TEM; the results are shown in Figure 2. For all OLC samples, agglomerates of mostly spherical particles were observed (Figure 2a,c,e). Other carbon nanostructures, for example, carbon nanofibers or nanotubes, were not observed. As shown in Figure 2b,d,f, HR-TEM imaging confirms the onion-like carbon structures in the as-synthesized OLC materials. For OLC-N, as typical of ND-derived OLC, ND residues were clearly visible as a result of the top-down processing from pyrolysis (Figure S3). Unlike OLC-N, both MOF-derived OLCs showed an irregular shape but with the constraint of interplanar distance between the graphitic layers in the shells. OLC-N had an average value of d spacing of 0.32 nm, which is slightly less than the commonly reported graphite interlayer distance (0.335 nm) [37]. The average d-spacing value of other OLC samples are 0.36 and 0.37 nm for OLC-B and OLC-M, respectively, which is larger than the previously reported value (0.33 nm) [26] and slightly smaller than our previous report (0.38 nm) [9].

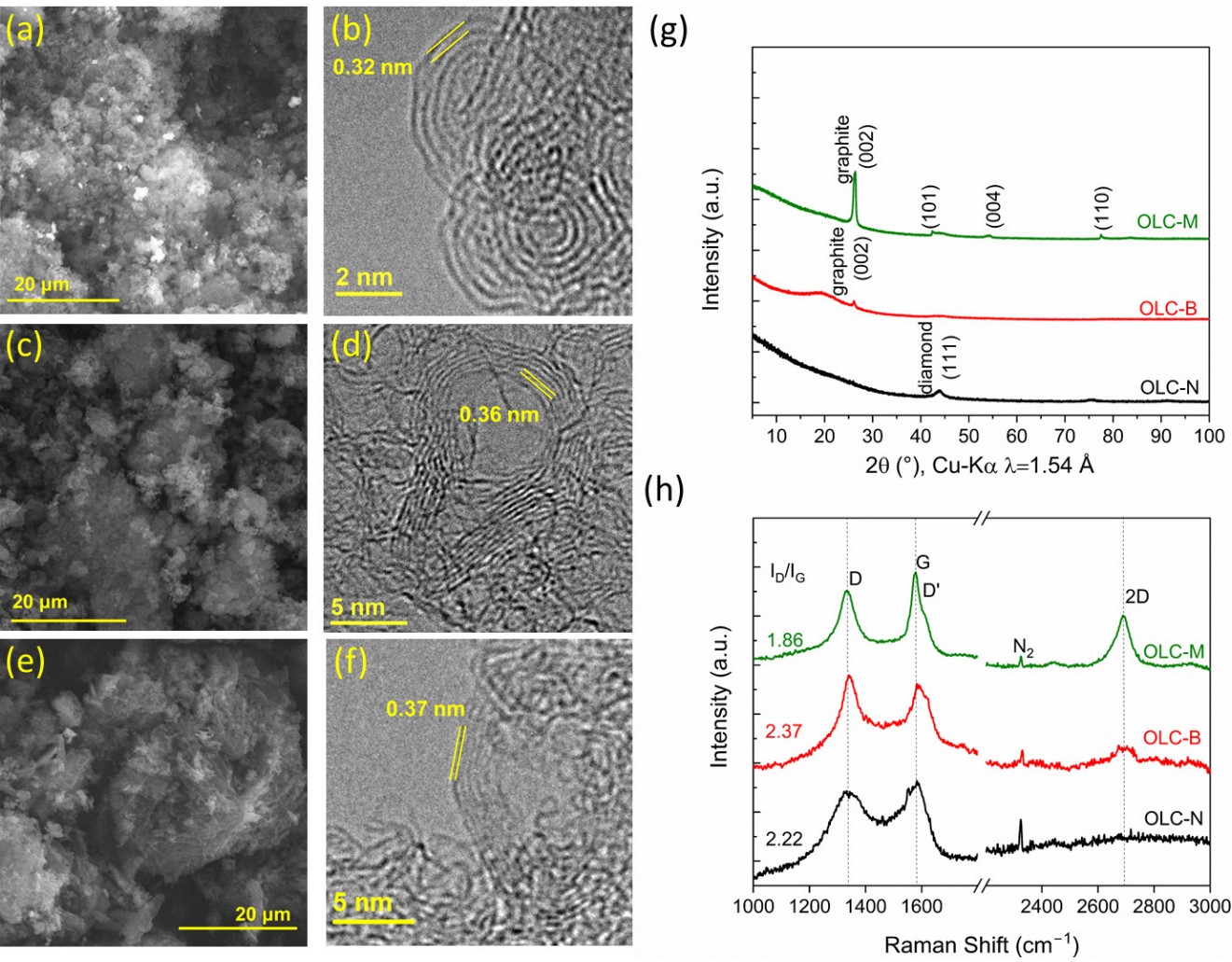

**Figure 2.** SEM images (SE detector) and HR-TEM images of (**a**,**b**) OLC-N, (**c**,**d**) OLC-B, and (**e**,**f**) OLC-M, (**g**) as well as XRD patterns and (**h**) Raman spectra for all OLC samples.

Figure 2g shows the XRD patterns for all OLC samples. There are two noticeable peaks at 43° and 76° detected in OLC-N, which were assigned to residual ND [31], in accordance with the ND residue observed in the high-resolution TEM image (Figure S3). There are no graphitic peaks in the XRD pattern of the OLC-N sample, which can be explained by the small particle size of produced OLCs, as shown in the HR-TEM image (Figure 2b). On the other hand, MOF-derived OLC samples OLC-B and OLC-M show a graphite (002) peak at 26° (PDF 41-1487). For OLC-B, a small broad reflection is observed around 20°, which can be associated with amorphous or disordered carbon. A more intensive crystalline (002) peak was observed for the OLC-M sample, as previously reported by Feng et al. [38], along with other graphitic peaks at 42° (101), 54° (004), and 77° (110) [39]. The intense graphite peaks likely suggest the formation of graphite-like carbon in the produced OLC samples, as the OLCs themselves are not expected to lead to sharp graphitic peaks in the XRD due to the nature and length scale of the crystallites. The higher crystallinity of the Fe-MIL100 precursor, as compared to disordered Fe-BTC (Figure S1a), is understood to lead to the formation of a significant portion of graphitic carbon [33]. As reported in our previous work, only a limited graphitic portion within the disordered OLC samples was concluded, based on small observed reflections [9]. It has been reported that for carbon nanomaterials with spherical morphology, an absence of sharp signals is observed in the XRD data [40,41]. Additionally, no respective peaks for the precursor MOFs are observed in the OLC-B and OLC-M samples, indicating a high conversion of the precursor.

Raman spectroscopy was performed to probe the nature of carbon in OLC samples; the spectra are shown in Figure 2h. All three samples show a pronounced D band at 1363 cm$^{-1}$ and a G band at 1587 cm$^{-1}$, along with additional shoulders/broadening. A detailed assignment of the shoulders and additional peaks for the three samples is presented in Figure S4. For OLC-N, aside from the D band and G band, pronounced intensity in the middle associated with an amorphous (A) band (1494 cm$^{-1}$) is observed, along with two additional peaks at 1324 and 1550 cm$^{-1}$, which may belong to the diamond band from ND, according to the Raman study of ND-to-OLC evolution [42]. With respect to OLC-B, four peaks can be assigned: D band (1340 cm$^{-1}$), amorphous (A) band (1494 cm$^{-1}$), G band (1585 cm$^{-1}$), and D$'$ band (1697 cm$^{-1}$). OLC-M is assigned the same D band (1332 cm$^{-1}$), A band (1491 cm$^{-1}$), G band (1576 cm$^{-1}$), and D$'$ band (1611 cm$^{-1}$). Substantial D-band signals are observed for all OLC samples, which is attributed to the presence of defects and disordered carbon in the graphitic structure. The A band represents amorphous carbon, which could be the result of interstitial disordered carbon, as suggested by Rouzaud [43], Jawhari [44], and Baranov [45]. The highest contribution from the A band in the Raman spectra is seen for OLC-N, whereas the lowest is associated with OLC-M, in line with the XRD data. Strong G-band signals associated with carbon sp$^2$ carbon of graphitic layers is observed for all samples. A G band of all OLCs, excluding OLC-N, is observed with a small shoulder D$'$, which, according to Gruen [46] and Klose [47], is attributed to the graphitic layer curvatures of the OLC shells. The absence of a D$'$ band for the OLC-N sample could be the result of low defect content from ND-derived OLC [48]. This low defect content might lead to poor electrochemical performance [49].

Additionally, the I$_D$/I$_G$ ratio of the D- and G-mode signals represents the graphitization degree in a material. The I$_D$/I$_G$ ratio values were evaluated to be 0.88, 1.23, and 0.78 for OLC-N, OLC-B, and OLC-M, respectively, as summarized in Table 1. The OLC-M sample has the lowest I$_D$/I$_G$, in line with the respective XRD pattern, showing the presence of crystalline graphite, as mentioned above. On the other hand, the OLC-N sample with a high I$_D$/I$_G$ ratio has no distinguishable crystalline peak in its XRD pattern, suggesting a large fraction of disorder.

**Table 1.** Summary of Raman, surface area, and pore volume details of OLC samples obtained by Raman spectroscopy and N$_2$ physisorption measurements.

| Sample | Raman I$_D$/I$_G$ | Specific Surface Area (m$^2$ g$^{-1}$) | Total Pore Volume (cm$^3$ g$^{-1}$) | Pore Volume$_{QSDFT}$ (cm$^3$ g$^{-1}$) | | |
| | | | | Micropores | Mesopores | Macropores |
|---|---|---|---|---|---|---|
| OLC-N | 0.88 | 505 | 1.37 | 0.04 | 1.33 | - |
| OLC-B | 1.23 | 594 | 0.59 | 0.13 | 0.46 | - |
| OLC-M | 0.78 | 180 | 0.23 | 0.02 | 0.19 | 0.01 |

An additional 2D band at 2689 cm$^{-1}$ is clearly observed for OLC-M. In the graphene literature, this feature is attributed to higher-order vibrational processes in graphitic layers [50]. A weak intensity or an absence of the 2D band, as observed for OLC-N and OLC-B, is reported to be the result of mixed sp$^2$-sp$^3$ hybridization, in addition to the presence of amorphous carbon [51]. Therefore, in line with the HR-TEM data, a considerable fraction of sp$^3$ carbon is likely present for both OLC-N and OLC-B. On the other hand, the distinguished 2D peak for OLC-M can also confirm the higher sp$^2$ fraction, which was detected as graphitic (002) from XRD.

N$_2$ physisorption experiments were performed to gain further insights into the morphology. The BET method was used to study the porosity and the specific surface area (Figure 3a,c,e). For all three OLC samples, QSDFT was used to evaluate the surface area and pore volume, with results summarized in Table 1. For OLC-N (Figure 3a), the specific surface area was determined to be 505 m$^2$ g$^{-1}$, with a total pore volume of 1.37 cm$^3$ g$^{-1}$. The isotherm has type-IVa characteristics, with a narrow hysteresis type H1 in the $p/p_0$ range

of 0.2–1, indicating a variety of pore size [52]. The OLC-B sample (Figure 3c), which was derived from Fe-BTC, has a higher specific surface area of 594 m$^2$ g$^{-1}$ but a low total pore volume of 0.59 cm$^3$ g$^{-1}$. The OLC-M sample (Figure 3e), which was synthesized from Fe-MIL100, produced the smallest specific surface area and total pore volume of 180 cm$^3$ g$^{-1}$ and 0.23 cm$^3$ g$^{-1}$, respectively. Physisorption experiments were also performed for the precursor MOFs and ND, with the data presented in Figure S5. For ND, an increase in surface area is observed after carbonization. In accordance with the literature, [33] Fe-MIL100 was found to have a higher surface area than that of Fe-BTC. However, the pyrolyzed carbons derived from that material, OLC-B and OLC-M, exhibited opposite results in terms of surface area and porosity, similar to the report by Feng and coworkers [38]. Similar to the OLC-N isotherm type, the MOF-route OLCs (OLC-B and OLC-M) also showed a type IVa isotherm with a mixture of type H2 and H3 hysteresis, as obtained for OLC derived from commercial MOF. A mainly H3 type hysteresis loop (possibly with a small mixture of H2(b)-type hysteresis at higher relative pressure ($p/p_0$)) is demonstrated by the closure of the loop around $p/p_0$~0.42. This is understood in the literature to signify capillary or advanced condensation in slit-like pores, along with a cavitation-like pore opening in the desorption process [47,53–55]. In contrast to OLC-N, MOF-derived OLCs show noticeable widening of the hysteresis loop as compared to precursor MOFs, likely resulting from the change in porosity due to inner voids.

Histograms of pore size distribution of all OLC samples are presented in Figure 3b,d,f. As mentioned with respect to the H1-type hysteresis of the OLC-N sample, unlike the other OLC samples, it shows a wide variety of pore size, from micro-size pores to the end of the measurement (35 nm). This wide size range may have been affected by the inhomogeneous size of its nanodiamond precursor. On the other hand, OLC-B (Figure 3d) showed a uniform mesoporous size, with the majority at ~3.75 nm. OLC-M (Figure 3f) showed major fraction of micropores and a small portion macro-sized pores, as summarized in Table 1. The macroporous fraction resulting from the enlargement of the micro-to-meso-size pores seems to be related to the OLC-M precursor as a result of the additionally elevated temperature processing during the initial autoclave step in contrast to OLC-B. This also might have affected the active surface area, as previously mentioned.

XPS was performed on the three OLC samples; the survey spectra are presented in Figure S6a,c,e. XPS elemental analysis (Table S1) confirms the high purity of the samples. OLC-N contains 97% carbon and a small amount of O (~0.8%), possibly from surface oxygen bonds resulting from air exposure. OLC-B and OLC-M have 95% and 99% of carbon contents, respectively. Minimal N and O were detected for the OLC-B sample, which may be residue from the starting material (Fe(NO$_3$)$_3$·9H$_2$O) during MOF synthesis. A Cl-content less than 1% was found to be the residual result of the etching process with HCl for the OLC-B sample. Surprisingly, the OLC-M sample only contains C and O, likely due to the different precursor, Fe-MIL100, with a higher-order structure and crystallinity, leading to improved metal-etching processes from the OLC particles than OLCs produced with Fe-BTC. XPS elemental analysis data are also in agreement with the bulk elemental analysis data obtained from ICP-OES (Table S1), especially in terms of C and O content, confirming the high purity of the OLC samples.

Fitting was undertaken for the high-resolution C 1s spectra of the samples to elucidate carbon hybridization, as shown in Figure S6b,d,f and summarized in Table S2. The sp$^2$ band (284 eV, along with a satellite at 290 eV) was found to have a fraction of 60% area for OLC-N, 70% for OLC-B, and 73% for OLC-M. Additionally, an sp$^3$ band was observed at 286 eV. The residual NDs contribute to the C 1s spectrum of OLC-N, as the reference measurement of the pure NDs (Figure S6b, inset) reveals a corresponding contribution of two species, sp$^3$ and defective sp$^3$ (287 eV), in agreement with the literature [56]. An sp$^2$ satellite, as well as C–O (287 eV), C=O (289 eV), and C–O=C (292 eV) contributions at higher binding energies, were also observed.

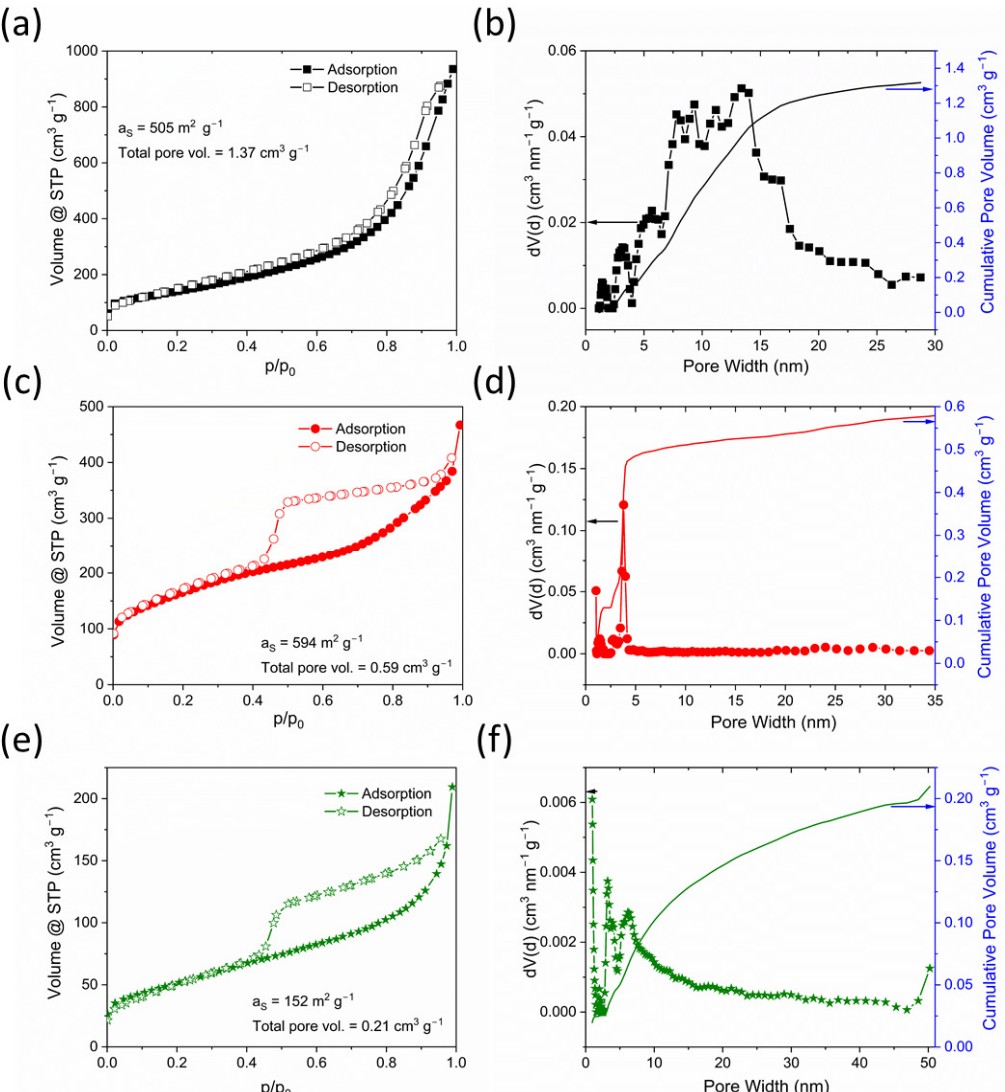

**Figure 3.** (**a**) N$_2$ physisorption isotherms and corresponding QSDFT pore volume and pore size distributions (PSD) of (**a**,**b**) OLC-N, (**c**,**d**) OLC-B, and (**e**,**f**) OLC-M.

Thus, depending on the precursor and synthesis conditions, MOF-based synthesis of OLC allows for high specific surface area with large pore volume. MOF-based synthesis represents a more cost-effective route to prepare OLC material with superior morphology in comparison to the ND-based synthesis route. Additionally, this synthesis route allows for control of the parameters of choice according to the precursor and the process conditions.

### 3.2. Electrochemical Performance in LIB Half Cells

The OLC samples were electrochemically tested as anodes (vs. Li/Li$^+$) in a 0.01 V $\leq U \leq$ 3 V potential range. Figure 4a,c,e presents GCPL voltage–capacity plots for the initial four cycles at 0.1 A g$^{-1}$ (~0.3 C rate based on the theoretical capacity of graphite, 372 mAh g$^{-1}$) of our OLC samples. As shown in Figure 4a, in the first discharge profile, OLC-N displays a plateau at 0.7 V, with a large, irreversible discharge capacity of ~1100 mAh g$^{-1}$ with respect to the discharge capacity in the second cycle, likely due to the solid–electrolyte interface (SEI) formation, as demonstrated by the extended plateau above 0.7 V. Despite the small surface area, the much smaller feature of OLCs in OLC-N may have led to higher activity toward side reactions. A reversible capacity of ~340 mAh g$^{-1}$ is observed in the initial stages. It was reported that the high fraction of disorder may be the reason for higher capacity of fullerene-like carbon nanostructures, in this case, OLC-N.

However, part of the disorder may cause Li-ions to be locally trapped and result a decrease in capacity [57].

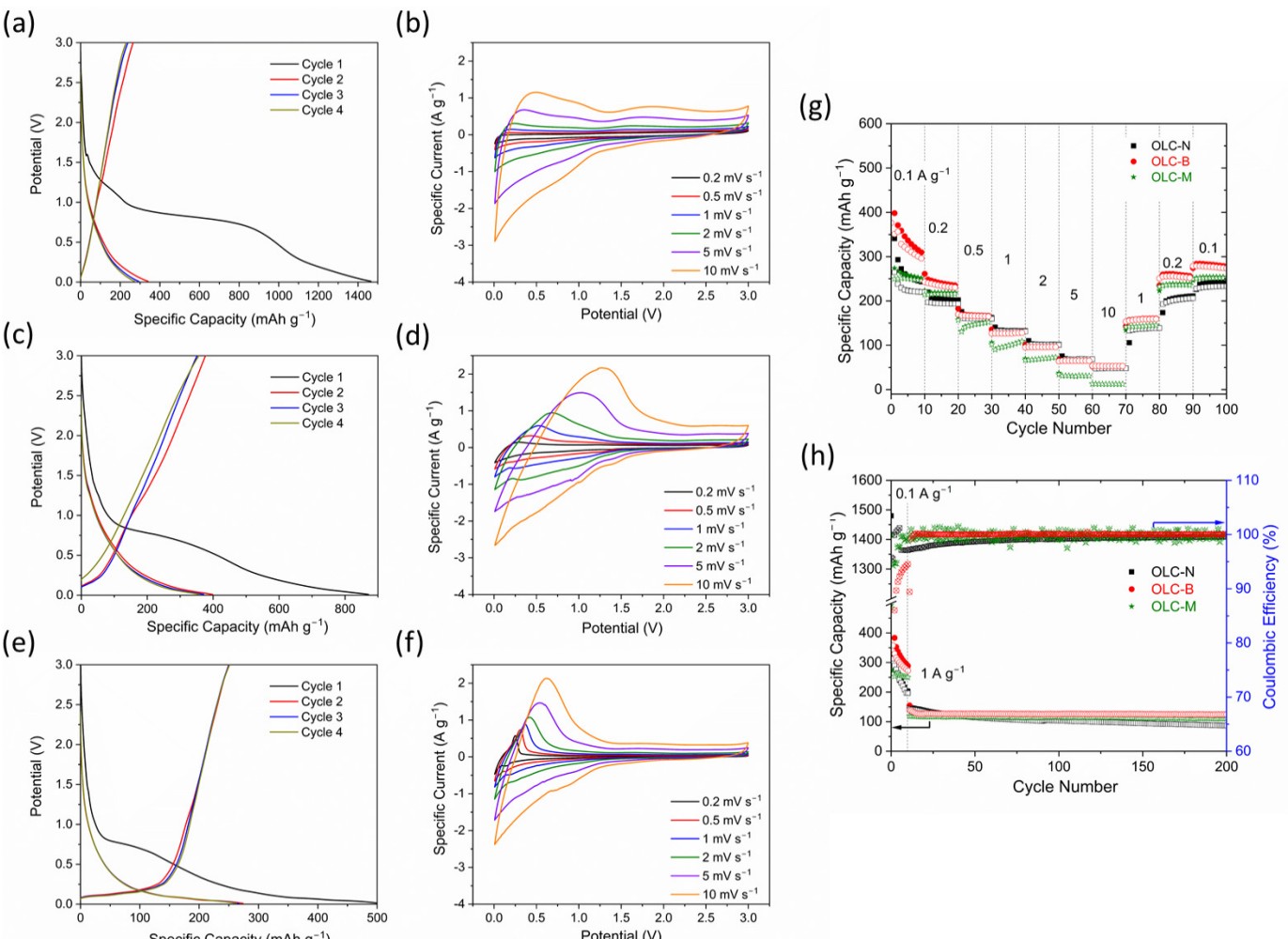

**Figure 4.** GCPL profile V vs. capacity of initial cycles at 0.1 A g$^{-1}$ and CV profiles at various scan rates for (**a**,**b**) OLC-N, (**c**,**d**) OLC-B, and (**e**,**f**) OLC-M, as well as GCPL (**g**) rate performance and (**h**) cycle performance in a LIB half-cell system.

The MOF-derived OLCs show a similar plateau at around 0.7 V, as shown in Figure 4c,e. Lower, albeit still significantly irreversible capacities of ~870 mAh g$^{-1}$ for OLC-B and ~500 mAh g$^{-1}$ for OLC-M (with respect to discharge capacity in 2nd cycle) were observed. Large surface area, as in the case of the MOF-derived OLC materials, was reported to cause additional entrapment of lithium, contributing to a high initial irreversible capacity [58] and leading to increased capacities. Although the irreversible capacities are lower as compared to OLC-N, OLC-B still has a higher initial reversible capacity of ~400 mAh g$^{-1}$ in contrast to ~270 mAh g$^{-1}$ for OLC-M. The voltage profiles for the two MOF-derived OLCs differ noticeably, with an additional pseudocapacitive-like capacity above 0.2 V for OLC-B. OLC-M shows a more graphite-like plateau below 0.2 V.

After an initial formation cycle, CV profiles were measured at varying scan rates (0.2–10 mV/s) and are presented in Figure 4b,d,f. In the potential window, the CV profiles of OLC-N (Figure 4b) display no sharp signals, indicating a gradual lithiation/delithiation with a shape similar to that of previously reported OLC materials [59] as also characteristic of porous carbon [60,61]. However, the CV measurement of OLCs from MOF showed a distinct profile. The OLC-M has a CV profile (Figure 4f) that resembles graphite [9] but with a smoother peak in terms of both cathodic and anodic sweep. This further supports

the presence of the significant amount of graphite in OLC-M that was detected by XRD and Raman measurements. At a low scan rate, the CV profile for OLC-B (Figure 4d) is similar to that of OLC-N. However, with an increase in the scan rate, the highest peak at anodic sweep also became larger and was shifted to a higher potential. A small amount of graphite may contribute to this shift, which was also observed in the XRD data. Therefore, the ordered nature of the MOF-precursor is understood to impact the graphitizability during carbonization. The rate performance of the OLC samples was evaluated at varying current densities from 0.1 to 10 A g$^{-1}$, as presented in Figure 4g. Although OLC-N shows a large irreversible capacity in the initial cycles, as previously discussed, the reversible capacity is the lowest up to 0.2 A g$^{-1}$. At the end of 10th cycle at 0.1 A g$^{-1}$, specific capacities of 243, 310, and 251 mAh g$^{-1}$ were obtained for OLC-N, OLC-B, and OLC-M, respectively. At a very high current rate of 10 A g$^{-1}$, a stable capacity of 48 and 53 mAh g$^{-1}$ was obtained for OLC-N and OLC-B, respectively, in contrast to 12 mAh g$^{-1}$ for OLC-M. OLC-N achieved better performance at high current density. The poor performance at a low current density is likely the result of low conductivity due to the presence of ND residue [62], along with a lack of defect sites, which are beneficial for Li-ion diffusion [58]. On the other hand, OLC-B achieves better rate performance compared to other samples due to a relatively high surface area, porosity, and number of defect sites, although still lower than that of commercial MOF-derived OLC [9]. OLC-M demonstrates the worst rate performance at every current density, likely due to the presence of a significant fraction of graphite, along with a small surface area, low porosity, and number of defect sites.

Figure 4h shows the cycling performance at 1 A g$^{-1}$ after initial formation cycles at 0.1 A g$^{-1}$. OLC-N exhibits decreasing values, with 87 mAh g$^{-1}$ by the end of 200 cycles, 60% capacity retention, and an average of 100% Coulombic efficiency. Both OLC-B and OLC-M show a relatively stable cycling at 1 A g$^{-1}$, as at the end of 200 cycles, OLC-M retained a specific capacity of 110 mAh g$^{-1}$, whereas OLC-B showed a specific capacity of 124 mAh g$^{-1}$. Moreover, the capacity retention of OLC-B is 90% and 93% for OLC-M, with an average Coulombic efficiency of 99.7% and 99.8%, respectively. Thus, as compared to nanodiamond-derived OLC, MOF-derived OLCs (OLC-B and OLC-M) deliver better electrochemical performance in LIB half cells.

EIS measurements were undertaken on the OLC samples to elucidate the contrasting electrochemical performance. EIS was performed for all samples in half-cell setup under open-circuit voltage (OCV) conditions before cycling and after 100 cycles in the charged state. Figure S7a,b shows the corresponding Nyquist plots. Two semicircles are observed for all samples after cycling as a result of SEI formation. The equivalent circuit models for fitting the impedance data are shown in Figure S7c, and the results are summarized in Table S3. Charge–transfer resistance or $R_{CT}$ of all OLC samples are significantly higher before cycling. In particular, OLC-N has the highest $R_{CT}$, indicating that it has an issue with conductivity. That could be expected from ND-derived OLC with ND residue (observed by HR-TEM), which is well-known for high resistivity [63]. Meanwhile, OLC-M has the lowest $R_{CT}$ before cycling. The presence of a large graphite fraction, as confirmed by various techniques, leads to high conductivity [64]. After cycling, the ionic pathways are established, and as expected, $R_{CT}$ is reduced for all OLC samples after 100 cycles. Both OLC-B and OLC-M show a much lower $R_{CT}$ than OLC-N after 100 cycles, suggesting good Li transference with cycling. On the other hand, the $R_{CT}$ of OLC-N is still markedly high, even after 100 cycles, possibly due to the presence of inactive ND residue. The $R_{SEI}$ values are quite similar for the three OLC samples.

According to physical and electrochemical characterization, a schematic of the three OLC samples is shown in Figure 5. OLC-N was synthesized by pyrolysis of ND at high temperature, leading to a solid core. The produced OLCs have an average particle size smaller than that of MOF-derived OLC, with a shorter interlayer distance and ND residue. For OLC-N, the average particle size is small, and the surface area is relatively high, although no crystalline graphite peak is observed in the XRD image. Although the CV profile shape is similar to that of mesoporous carbon, the charge storage could not reach its

maximum potential due to a solid core and ND residue with high charge–transfer resistance. OLC-B was produced via pyrolysis of self-made Fe-BTC. Starting with a heat-free synthesis process of Fe-BTC, the produced OLC has a significant fraction of amorphous content and defect sites, along with a small portion of graphite side product. Its particle size is larger than that of OLC-N; however, considering that there are inner voids after the Fe etching process, which are accessible due to the presence of many defect sites, the measured surface area by $N_2$ physisorption is the largest among three samples, along with its porosity. Therefore, the best electrochemical performance is associated with OLC-B, attributed to a high surface area and porosity, varied defect sites [65], and accessible inner voids [66]. On the other hand, OLC-M was prepared with self-made Fe-MIL100, which was highly crystalline. As a result, OLC-M has a high fraction of graphite side product, along with a low amorphous content, leading to a much smaller surface area and porosity in OLC-M as compared to OLC-B and OLC-N. Although the OLCs in OLC-M have an accessible inner void space, small surface area, and low porosity, a large fraction of graphite impacts the half-cell performance, especially the rate capability.

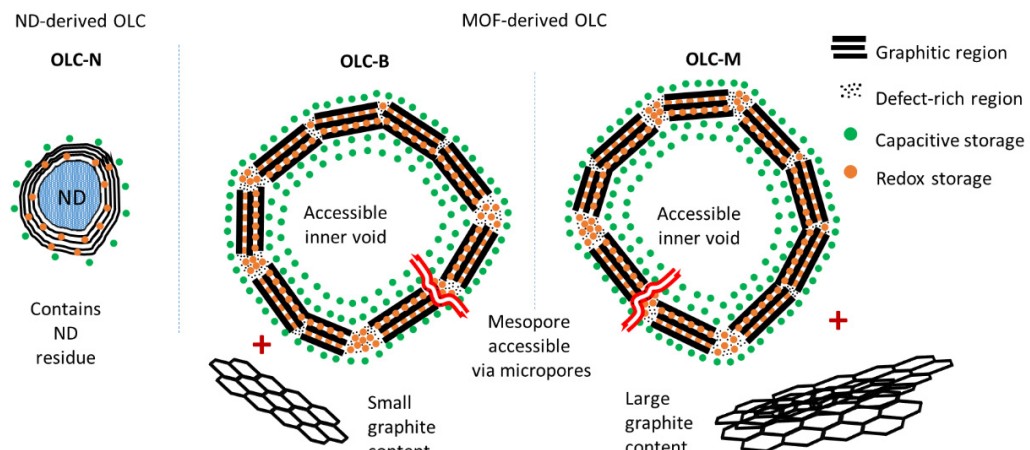

**Figure 5.** Schematic representation of the OLCs produced from nanodiamond and MOFs.

### 3.3. Electrochemical Performance in Full-Cell LICs

OLC anodes were paired with an activated carbon (AC) cathode for electrochemical testing of full-cell LICs. The electrochemical measurements for the LIC system were performed in a voltage window of 2 V $\leq U \leq$ 4 V. Figure S8a,c,e shows the CV profiles for the OLC-N//AC, OLC-B//AC, and OLC-M//AC LIC cells, respectively. The measurements were performed at scan rates in the range of 0.2 to 10 mV/s. As expected, CV profiles for all three LICs demonstrate the shape difference with respect to conventional supercapacitors, which gradually deviate from the ideal rectangular shape with increasing scan rates. As with asymmetric supercapacitors, this phenomenon is caused by pairing of different electrode materials in LICs [67]. Additionally, OLC-N//AC and OLC-M//AC showed a larger deviation at higher potentials (3 V $\leq U$).

The rate capability of the full-cell LIC systems was evaluated by galvanostatic cycling at current densities in the range of 0.1 to 10 A g$^{-1}$. As shown in Figure 6a, at every recorded current rate up to 10 A g$^{-1}$, OLC-B//AC and OLC-M//AC delivered higher specific capacitance as compared to OLC-N//AC. At a low current of 0.1 A g$^{-1}$, OLC-N//AC delivered a specific capacitance of 67 F g$^{-1}$, whereas OLC-B//AC and OLC-M//AC achieved specific capacitances of 178 and 111 F g$^{-1}$, respectively. At a high current of 10 A g$^{-1}$, a significant drop in specific capacitance is observed for all OLC-LICs; OLC-N//AC, 12 F g$^{-1}$; OLC-B//AC, 68 F g$^{-1}$; and OLC-M//AC, 39 F g$^{-1}$. OLC-B//AC showed the best initial rate capability, and the rate capability of OLC-M//AC was also significantly better than that of OLC-N//AC. For practical purposes, it is also important to understand the rate capability after sustained cycling. Therefore, after the initial rate performance measurement,

all the LICs were cycled at 1 A g$^{-1}$ for 1000 cycles. Rate performance was evaluated for all LICs, with data shown in Figure 6b. At 0.1 A g$^{-1}$ current density, OLC-N//AC has 48 F g$^{-1}$ specific capacitance, whereas OLC-B//AC and OLC-M//AC show much higher specific capacitances of 117 F g$^{-1}$ and 73 F g$^{-1}$, respectively. When the current density is increased to 10 A g$^{-1}$, OLC-N//AC only shows a specific capacitance of 15 F g$^{-1}$, with OLC-M//AC also showing a low specific capacitance of 25 F g$^{-1}$, whereas OLC-B//AC maintains a specific capacitance of 71 F g$^{-1}$. A deterioration in overall specific capacitances at varying rates for is obvious for all three LICs. However, OLC-B//AC maintains sufficiently high capacitances, along with good rate capability, whereas the performances of both OLC-N//AC and OLC-M//AC are similarly reduced.

The galvanostatic charge–discharge curves in Figure S8b,d,f show the potential vs. time profile for OLC-N, OLC-B, and OLC-M, respectively. Figure S8b,f shows that the profile of OLC-N//AC and OLC-B//AC has a non-linear curve at a 0.1 A g$^{-1}$ current density, whereas OLC-M has an almost linear and symmetrical response at the same current density, indicating a deviation trend in the order of OLC-M → OLC-B → OLC-N. The minimal deviation from the linear slope of the GCPL profile could also be the result of overlapping effects between the non-Faradaic reaction of the cathode and the Faradaic reaction of the anode [67]. Additionally, the IR drop graph in Figure 6c shows that OLC-N has the highest value with increasing current density, followed by OLC-B, whereas OLC-M has the lowest value. It has been discussed in literature that a lower IR drop indicates an improved electrical conductivity due to quick electron transfer [68]. Accordingly, the trend of the three OLC LICs is also in agreement with EIS measurements in half-cell LIBs, as discussed earlier.

The rate performances of all three LIC systems (OLC-N//AC, OLC-B//AC, and OLC-M//AC) in the voltage window of 2 V $\leq U \leq$ 4 V were used to calculate the energy density and power density at varying current densities (details in Supplementary Materials). The obtained values are summarized as a Ragone plot in Figure 6d. Additionally, the data from commercial Fe-BTC-derived OLC and graphite LIC systems from our previous study are shown for comparison [9]. As observed with respect to the LIC rate performance (Figure 6a), OLC-N//AC shows an inferior performance to that of both OLC-B and OLC-M LIC systems. At the maximum energy density of 91 Wh kg$^{-1}$, OLC-N//AC has a 152 W kg$^{-1}$ power density, whereas OLC-B//AC and OLC-M//AC have power densities of 243 Wh kg$^{-1}$ (211 W kg$^{-1}$) and 162 Wh kg$^{-1}$ (223 W kg$^{-1}$), respectively. On the other hand, at a maximum power density of 15,189 W kg$^{-1}$, OLC-N//AC delivers 12 Wh kg$^{-1}$, compared to OLC-B//AC, with 20,149 W kg$^{-1}$ (66 Wh kg$^{-1}$), and OLC-N//AC, with 24,599 W kg$^{-1}$ (48 Wh kg$^{-1}$). Compared to commercial graphite LIC [9], LICs with MOF-derived OLCs demonstrate outstanding energy and power densities, whereas OLC-N LIC shows the worst performance. The energy density of OLC-M LIC is lower than that of OLC-B LIC. Moreover, the energy and power density for LIC containing OLC-B synthesized from the self-made Fe-BTC precursor, are better than those of LIC with commercial Fe-BTC-derived OLC [9].

Furthermore, after initial formation cycling for 10 cycles at 0.1 A g$^{-1}$, long-term cycling at a high current rate of 1 A g$^{-1}$ was performed for all OLC LIC systems (Figure 6e). After formation cycling, a current rate of 1 A g$^{-1}$ OLC-N//AC showed a specific capacitance of 79 F g$^{-1}$, in comparison to much higher specific capacitances of 156 F g$^{-1}$ and 106 F g$^{-1}$ for OLC-B//AC and OLC-M//AC, respectively. After a further 10,000 cycles at a current density of 1 A g$^{-1}$, OLC-N//AC delivered only 33 F g$^{-1}$, with 42% capacitance retention. On the other hand, MOF-derived OLC LIC systems achieved much better cycle performance, with OLC-B//AC reaching 122 F g$^{-1}$ with 78% capacitance retention and OLC-M//AC showing 86 F g$^{-1}$ with 81% capacitance retention. Despite the variation in capacitance retention for all of OLC LIC systems, high average Coulombic efficiencies of ~100% over 10,000 cycles were obtained for MOF-derived OLCs, with 99% Coulombic efficiency for ND-derived OLC. As summarized in Table S4, both the MOF-derived OLC-based LICs demonstrate an overall better combination of maximum energy density, maximum

power density, and cycle life compared to the state-of-the-art LIC literature reporting on carbon-based materials as anodes.

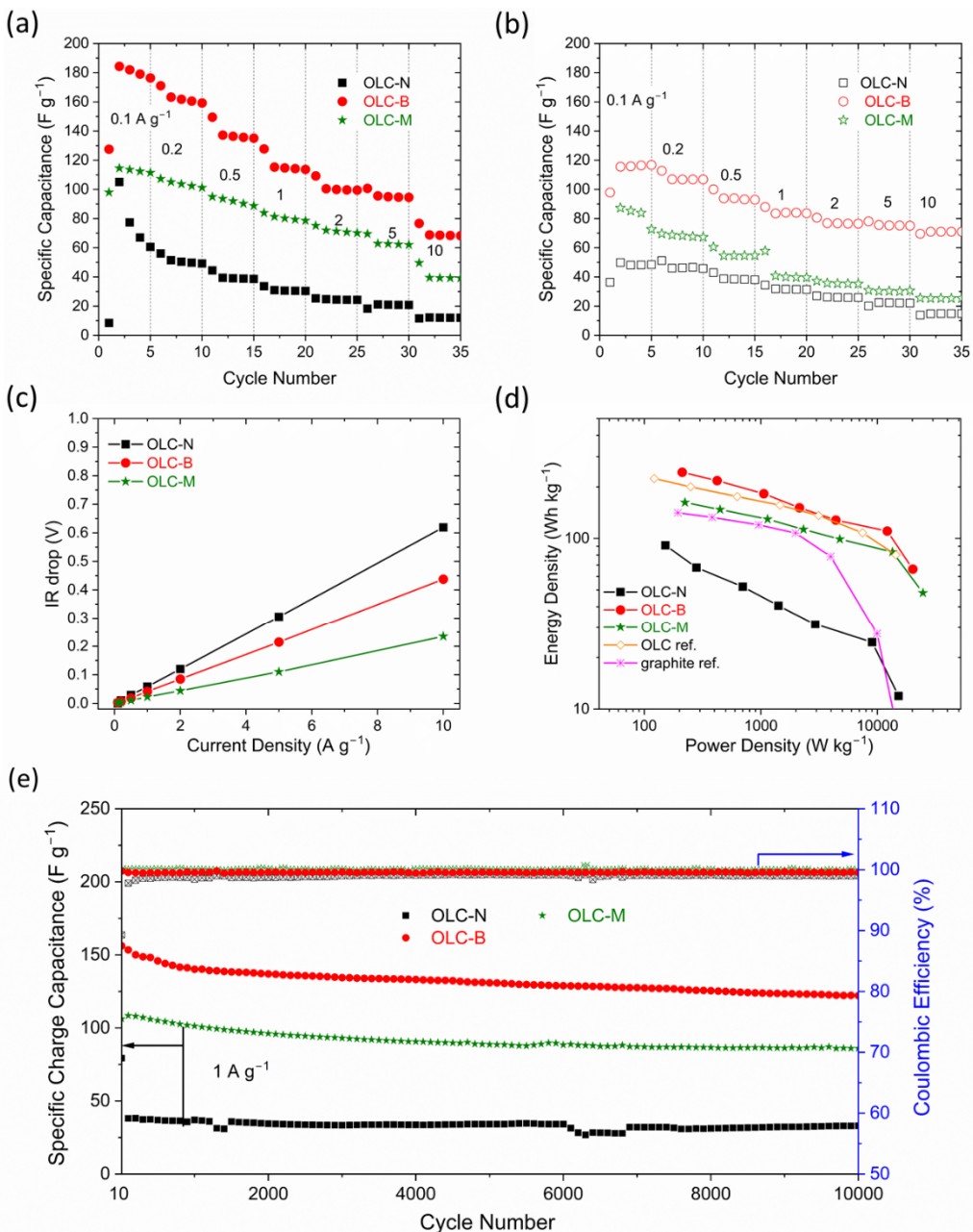

**Figure 6.** Electrochemical performance of LICs with varying OLCs (**a**), GCPL rate performance (current rates in A$^{-1}$) of fresh LIC cells, and (**b**) GCPL rate performance (current rates in A g$^{-1}$) after 1000 cycles at 1 A g$^{-1}$ for the same cells in (**a**). (**c**) IR drop at every current density in (**a**), (**d**) Ragone plot (normalized to active material in all electrodes) in comparison with the results of our previous study, and (**e**) GCPL cycling performance at a high current rate of 1 A g$^{-1}$ (normalized to active material in anodes).

The ND-derived OLC with less-conductive ND residue, resulted in poor rate performance, even with a relatively high surface area. It is clear that nanodiamond-derived OLC is not suitable for application in LIC systems. Nevertheless, some studies have used a ND-derived OLC material as a template or in a composite with metallic compounds [22,23]. On the other hand, the MOF-derived OLCs, OLC-B and OLC-M, have the advantage of numerous defect sites and accessible inner voids, enabling better electrochemical performance.

The performance of OLC-M was affected by the presence of a large fraction of graphite, resulting in reduced power densities relative to the OLC-B LIC system.

Through using two types of MOF precursors with the same base units, we obtained an understanding of the impact of precursor structure and morphology on the resulting OLC material, as well as its impact on electrochemical performance. Moreover, the MOF-pyrolysis route is straightforward, yielding high-purity materials and allowing for parametric control of the structure, morphology, and dopants through the use of different MOFs/additives and modifications in system atmosphere and pressure. Thus, as compared to the OLCs obtained via the nanodiamond route, MOF-derived OLCs offer excellent performance, both in LIB half cells and full-cell LIC systems.

## 4. Conclusions

We investigated OLC materials synthesized via two routes, pyrolysis of conventional NDs and carbonization of MOFs, with respect to their potential application in hybrid capacitors. The OLCs were evaluated in terms of their structure and morphology, as well as their electrochemical performance in LIB half cells, as well as in LIC full cells paired with commercial activated carbon. The precursor structure and synthesis temperature strongly impacted the porosity surface area and defect concentration. Even for the MOF route, the difference in the precursor MOF structure led to OLC from Fe-BTC (OLC-B) to have a larger surface area and porosity as compared to OLC from Fe-MIL100 (OLC-M). On the other hand, OLC from nanodiamond (OLC-N) showed the highest pore volume and a high surface area comparable to that of OLC-B. However, owing to its synthesis route, OLC-N demonstrated much more compact nature of the individual OLCs and lower defect concentration in the shells, leading to the worst electrochemical performance. Given the aforementioned merits, the OLC-B LIC system delivers a maximum energy density of 243 Wh kg$^{-1}$ with a maximum power density of 20,149 W kg$^{-1}$ and a good cycling stability with 78% capacitance retention after 10,000 cycles. Hence, the MOF-derived OLC demonstrates potential for application in practical LICs. Moreover, the facile MOF-derived synthesis route should enable easy industrial application.

**Supplementary Materials:** The following supporting information can be downloaded at: https://www.mdpi.com/article/10.3390/batteries8100160/s1; Supporting information contains the calculation methods of capacity and capacitance [69]; Figure S1: (a) XRD patterns of ND, Fe-BTC, and Fe-MIL100 and FT-IR spectra of (b) Fe-BTC, Fe-MIL100, commercial Basolite F300, and (c) ND.; Figure S2: Thermogravimetric analysis of (a) Fe-BTC and (b) Fe-MIL100.; Figure S3: HR-TEM image of OLC-N at a different spot, emphasizing the ND residue.; Figure S4: Raman peak assignment for (a) OLC-N, (b) OLC-B, and (c) OLC-M.; Figure S5: Isotherms from N$_2$ physisorption experiments for (a) ND, (b) Fe-BTC, and (c) Fe-MIL100.; Figure S6: XP spectra survey and C 1s fitting for (a,b) OLC-N, (c,d) OLC-B, and (e,f) OLC-M; the inset of (b) shows the fitted C 1s spectrum for pure nanodiamond.; Figure S7: Nyquist plots from EIS measurements on half-cell LIB (a) before and (b) after 100 cycles for OLC samples, along with the fitted curves based on (c) equivalent circuit models used for before and after cycling data; Figure S8: CV profiles at various scan rates and GCPL voltage vs. time profiles at various current densities of for (a,b) OLC-N//AC, (c,d) OLC-B//AC, and (e,f) OLC-M//AC in an LIC system (normalized to the mass of active material in anode).; Table S1: Elemental analysis of OLC samples from XPS and ICP-OES.; Table S2: Summary of fitting of the high-resolution C 1s spectra for OLC-N, OLC-B, and OLC-M samples.; Table S3: Summary of fitting of EIS data for half-cell LIB of OLC samples before and after 100 cycles.; Table S4: Comparison of several reported LIC systems with carbonaceous anodes [9,24,70–81].

**Author Contributions:** Conceptualization, A.D.C.P., D.M. and A.O.; methodology, A.D.C.P., D.M. and A.O.; validation, A.D.C.P., D.M. and A.O.; formal analysis, A.D.C.P. and M.H.; investigation, A.D.C.P., L.D., I.G.G.-M. and M.H.; resources, D.M.; data curation, A.D.C.P., L.D., I.G.G.-M. and M.H.; writing—original draft preparation, A.D.C.P.; writing—review and editing, A.D.C.P., D.M. and A.O.; visualization, A.D.C.P., D.M. and A.O.; supervision, A.O., D.M. and K.N.; project administration, A.O. and D.M. All authors have read and agreed to the published version of the manuscript.

**Funding:** A.D.C.P. would like to acknowledge the doctoral grant from Deutscher Akademischer Austauschdienst (DAAD). A.O. would like to thank the Federal Ministry of Education and Research (BMBF) for financial support under grant no. 03XP0254D (Cluster of Competence for Battery Materials ExcellMatBat: KaSiLi project).

**Data Availability Statement:** The data presented in this study are available from the corresponding authors upon reasonable request.

**Acknowledgments:** The authors would like to thank A. Voß, A. Voidel, and H. Bußkamp for the ICP/CGHE measurements and B. Bartusch for the TGA (all from IFW Dresden).

**Conflicts of Interest:** The authors declare no conflict of interest.

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
