# Peer review of "Comparative Study of Onion-like Carbons Prepared from Different Synthesis Routes towards Li-Ion Capacitor Application"

_batteries, doi:10.3390/batteries8100160_

Round 1

Reviewer 1 Report

         The authors synthesized two MOF-derived nanoporous carbons and the conventional nanodiamond-derived porous carbon and utilized them in Li-ion capacitors. The idea is interesting, the findings are promising, and the work is well organized with lots of experimental data to support most conclusions. Thus, I think this is a nice work to be published. I suggest minor revision for the authors to address the following issues before publication.

1.      Nitrogen adsorption-desorption isotherms of the two MOFs and pristine nanodiamond should be reported and compared with the data shown in Figure 3. Such data can also show the porosity of both MOFs to verify that the MOF synthesis was successful.

2.      In additional to morphology and porosity, another important factor for the use of nanoporous carbon in electrochemical applications is electrical conductivity. Thus, it is highly suggested to measure the bulk electrical conductivity of all the three materials (OLC-B, OLC-M and OLC-N). This experiment can be done by simply pelletizing the bulk powder and perform either the two-probe or four-probe I-V measurements.

3.      The OLC-B material performs the best here, but from Table S1, there is a minor amount of Fe remaining in the OLC-B material. Is there any effect from the residual Fe on the resulting electrochemical performances? I would suggest the authors to add some comments in the manuscript regarding this point.

4.      The authors should add the purity of each chemical used here in section 2.1.

Reviewer 2 Report

The article described a comparison of several carbon-based materials for LICs. The field of battery materials is generally lacking such comparisons, therefore, such work is greatly appreciated by the community.

However, few minor issues should be corrected:
1. the authors should provide a clear explanation of the studies samples (such description is given on the page 13, instead of beginning of the section 3).
2. The experimental part described the synthesis, however, the molar amounts are not provided, neither is chemical yield.
3. the authors need to explain the origin of rather low irreversible capacity loss during half-cell experiments for OLC-B - as it has the highest surface area.
4. The trends in rate performances in half-cell and full-cells are quite different (almost no differences between the materials in half-cells). Can the authors explain why?
5. I would suggest the authors not to use the words such as "superior", when describing their materials.

Reviewer 3 Report

The authors prepared  Onion-like Carbons Prepared from Different Synthesis Routes Towards Li-ion Capacitor Application, which exhibited a energy density of 243 Wh/kg and power density of 20149 W/kg. There are some major issues in this manuscript blocking the full acceptance of this work to be published by batteries journal. The authors should response the following questions:

1. Authors claim they calculated the capacitive and diffusive contributions according to Dunn method. Then where is the shaded graph of the capacitive/diffusive contributions with respect to total contributions?

2. In the XRD pattern the peaks are not  indexed and there is no standard JCPDS card information for comparison.

3 . Comparison of reported LIC systems with carbonaceous anodes table  should include more literature review.

4. Post stability data should be provided which includes, EIS analysis, SEM images, also valence state.

Reviewer 4 Report

In the proposed manuscript “Comparative Study of Onion-like Carbons Prepared from Different Synthesis Routes Towards Li-ion Capacitor Application” the authors made onion-like carbons by three different ways and studied it. The work is interest and well described.

Just few moments can be improved:

1.      “BTC” is not described. The same for “MIL-100”.

2.      In line 215: “Zeiger et al., 2015” is mentioned, but it is not included in the list of Refs. (only Zeiger et al., 2016) Please, check it.

3.      Any reason why for anode slurry was done as 85:5:10 but for cathode as 70:10:20? Also about the difference between mass loading: ~1.2 mg/cm2 for anodes and ~4 mg/cm2 for cathode.

4.      It will look better if “1/cm” will be changed to “cm-1” (related to Raman).

5.      Please, add formulas that were used for the calculations in the current work.

6.      Table for the comparison “Table S4” can be included to the main text.
